Not so cryptic–differences between mating calls of Hyla arborea and Hyla orientalis from Bulgaria

http://orcid.org/0000-0002-2546-6590 Lukanov Simeon simeon_lukanov@abv.bg
Institute of Biodiversity and Ecosystem Research, Bulgarian Academy of Sciensces , Sofia , Bulgaria
Manjarrez Javier
Electronic publication date: 2024 Jun 25
Publication date: 2024
Volume: 12
Electronic Location ID: e17574
Received 2024 Mar 20; Accepted 2024 May 24
Copyright: © 2024 Lukanov
Copyright year: 2024
Copyright holder: Lukanov
License: This is an open access article distributed under the terms of the Creative Commons Attribution License, which permits unrestricted use, distribution, reproduction and adaptation in any medium and for any purpose provided that it is properly attributed. For attribution, the original author(s), title, publication source (PeerJ) and either DOI or URL of the article must be cited.
License URL: https://creativecommons.org/licenses/by/4.0/

Keywords: Bioacoustics, Call parameters, Sister species, Taxonomy, Treefrogs

Funding: National Science Fund of Bulgaria under KP-06-N31/11 This work was supported by the National Science Fund of Bulgaria under Grant contract № KP-06-N31/11 from 11.12.2019. The funders had no role in study design, data collection and analysis, decision to publish, or preparation of the manuscript.

==============================
Anurans are among the most vocally active vertebrate animals and emit calls with different functions. In order to attract a mate, during the breeding season male frogs produce mating calls which have species-specific structure and parameters, and have been successfully used to resolve issues in taxonomy and phylogenetic relations. This is particularly useful when closely related taxa are concerned, as many species are morphologically almost identical, but still their status is well-supported by molecular and genetic data, suggesting the existence of mechanisms for reproductive isolation. Such is the case for treefrogs from the Hyla arborea group, which are now recognized as several distinct species. The present study aims to establish differences in call parameters between the European tree frog, Hyla arborea, and the Eastern tree frog, Hyla orientalis, which both occur on the territory of Bulgaria. Using autonomous audio loggers, calls from six sites (three in the range of H. arborea and three in the range of H. orientalis) were recorded between 7 p.m. and 12 a.m. during the breeding season in 2020–2023. The following parameters in a total of 390 mating calls were analyzed: call count, pulse count, call series duration, call period, peak (dominant) frequency, entropy. Results indicated that sites formed two distinct groups, which corresponded to the known distribution ranges of H. arborea and H. orientalis. The first two components of the PCA explained 71% of the total variance, with variables call count, call series duration, peak frequency and entropy being most important for differentiation between the sites. This study presents the first attempt to differentiate between the calls of these two sister taxa, which both fall within the “short-call treefrogs” group, and results are discussed in terms of known data for mating calls in Hyla sp., as well as limitations and future perspectives.

Introduction

Anurans are the most vocally active amphibians and can emit distinct calls with differing functions in their social behaviour (e.g., to mark territory, to attract mate, to indicate danger, etc.). Mating (or advertisement) calls, which are produced by males to attract a mate during the breeding season, can be very specific and can serve to distinguish between different species (Ryan, 2001). As a result, the examination of vocalization is widely employed to clarify taxonomic and phylogenetic issues. Moreover, as acoustic interference between heterospecific males can lead to temporal or spectral separation, it is considered a significant feature for identifying species (Schneider & Sinsch, 2007; Köhler et al., 2017).

Acoustic signals are of key importance for anuran mate recognition system, and the effects of sexual selection can manifest as either stabilizing or directional (Castellano et al., 2002). A number of studies have demonstrated that different call characteristics may experience distinct selective pressures (e.g., Bee, 2007; Höbel, 2015; Vélez & Guajardo, 2021). According to Gerhardt (1992) anuran mating calls encode multiple messages of both species identity and mate quality, with some call characteristics (e.g., call duration) often under directional preferences, while others (e.g., call frequency) under stabilizing selection. Characteristics under stabilizing preference are much more static than those under directional preference, which tend to be more dynamic (Castellano & Giacoma, 1998).

Hyla is the only genus from the large Hylidae family that occurs outside of the New World, and it is widespread over Eurasia and parts of Northern Africa. Members of the H. arborea species complex are the only representatives for mainland Europe, and their taxonomy has undergone rapid and dynamic changes in recent decades–from a single species in the 1970s to nine species today (Gvoždík et al., 2015). The European tree frog, Hyla arborea, is distributed from the Southern Balkans to North-Western Europe, and the Eastern tree frog, Hyla orientalis-from Anatolia to North-Eastern Europe. Their contact zone runs from North-Eastern Greece to the Central Balkans along the Carpathian chain, and further north across lowland Poland along the Vistula River (Stöck et al., 2012). H. orientalis was separated as a species from the group of H. arborea relatively recently–in 2008 by Stöck et al. (2008) based on mitochondrial and nuclear DNA data. According to molecular data, most treefrog populations on the territory of Bulgaria belong to the H. orientalis taxon, while H. arborea is present in the region of the Struma river basin (Dufresnes et al., 2015). Although these taxa are thought to have diverged during the Mio-Pliocene (~5 Mya), and are currently accepted as separate species (Speybroeck et al., 2020), they are morphologically so similar that they cannot be distinguished based on external characteristics.

Bulgaria is the range limit for H. orientalis and available studies demonstrate that the border between the two species is well established and historically constant, but there is no further evidence on the precise mechanisms for reproductive isolation that maintain it. The present study is the first attempt to differentiate the two species based on their mating calls. I tested the hypothesis that calls from localities within the established range of the respective species will differ in terms of their spectral and temporal characteristics.

Materials and Methods

Study sites

Recordings from six sites were used in the analyses–three from South-West Bulgaria (area with H. arborea) and three from the rest of the country (area with H. orientalis) (Fig. 1). All sites are temporary water ponds with depth of 0.5–1.5 m, underwater vegetation and banks overgrown with reed or bulrush.

Figure 1 Geographical location of study sites in Bulgaria.

The region within the range of H. arborea is in light green, the rest of the country is occupied by H. orientalis (following Dufresnes et al., 2015). Source credit: map created with ArcGIS.

Call recordings

Recordings were made during the breeding season (April–June) in 2020–2023. AudioMoth acoustic loggers were positioned and regularly checked at all sites (one logger per site), recording in WAV-PCM format with sample rate set at 32 kHz and 24-bit resolution. For one site (Livada) recordings were made with Wildlife acoustics SongMeter SM4 with the same settings. All frogs were calling from the water and recordings used for the analyses were made between 7 p.m. and 12 a.m., at ambient temperatures of 18–20 °C. For the site Livada, daily air temperature for the study period was taken from the metadata of the recordings, and for the other sites data were collected from the nearest automatic weather stations of the Bulgarian National Institute of Meteorology and Hydrology (available in Bulgarian at https://www.stringmeteo.com/). A total of 390 mating calls were used for the analyses: 51 from Arkutino, 60 from Dobrusha, 61 from Livada, 54 from Plovdiv, 82 from Rupite and 82 from Ruzhdak. As the difference in intensity and tonality of mating calls allows for limited individual recognition (Crovetto, Salvidio & Costa, 2019), no more than three calls from the same animal were used from each site. All mating calls used in the analysis were of individual male frogs, calling near the audio recorder, with low level of ambient noise and no chorus in the background. Overlapping calls of two or more frogs, calls that were part of a chorus, or too distant calls were not considered for the analyses. All research was carried out in accordance of Bulgarian Ministry of Environment and Water permit No 861/13.01.2021.

Call parameters

The definitions of call structure followed Castellano et al. (2002), who define the advertisement call of European treefrogs as a call group (series) comprised of individual calls, each call consisting of pulses. The following call parameters were analysed: Call count (number of calls in a call series), Pulse count (number of pulses in a call), Call series duration (time between first and last call of a call series, measured in seconds), Call period (time between the beginning of a call and the end of the interval after this call, measured in seconds), Peak (dominant) frequency (frequency of maximum power, measured in Hertz), Entropy (ratio of the geometric mean to the arithmetic mean of the spectrum). Measurements were taken using the Pulse Train Analysis tool in Avisoft SASLab Pro v. 5.2.14 (Avisoft Bioacoustics, Nordbahn, Germany), with pulse detection by peak search with hysteresis, rectification and exponential decay, and Fast Fourier Transform (FFT) size of 1,024. Values for all parameters were copied directly from the pulse train results, except for Call period, which was calculated as a sum of the call and interval durations. Spectrograms and oscillograms of the calls were made with the software SoundRuler v. 0.9.6.0. (Gridi-Papp, 2003–2007), again with FFT size of 1,024.

Statistical analyses

To account for pseudoreplication, mean values for individual call series (n = 390) and individual males (n = 160) were calculated, and the latter were used for the subsequent analyses. To ensure that each variable contributes equally to the analyses and to prevent variables with larger scales from dominating the results, all variables were standardized by means of z-score normalization. Multivariate Analysis of Covariance (MANCOVA) with “Site” as a grouping variable was performed in order to examine whether there were significant differences in the frog call parameters among different sites, followed by least significant difference (LSD) tests between all sites. A principal component analysis (PCA) was used to establish which variables contributed most to the distinction among the sites, and for further processing, I used the variables with loading values greater than 0.5. The first two components of the PCA were used in a scatterplot to demonstrate the distribution of mating calls from the different sites. Linear discriminant analysis (LDA) was performed to classify sites based on acoustic call parameters, the model was fitted with four discriminant functions and a k-fold cross-validation was executed. Posterior probabilities were extracted from the LDA results and Euclidean distances were calculated between observations in the posterior probabilities space. For each of the six sites, the average distance to observations from other sites was computed.

All statistical analyses were carried out in R version 4.1.3 (R Core Team, 2021), and the chosen level for statistical significance was p < 0.05.

Results

The values of all analyzed parameters are presented in Table 1.

Table 1 Values for the analysed parameters from all six study sites.

Data is presented as Min–Max (Mean ± SD).

	Within the range of Hyla arborea	
Livada	Rupite	Ruzhdak	
Call count	10–51 (23 ± 10.3)	9–32 (21 ± 5.6)	10–32 (18 ± 5.4)	
Pulse count	5–9 (6 ± 0.6)	5–8 (6 ± 0.6)	5–9 (6 ± 0.8)	
Call series duration (s)	1.8–12.2 (4.7 ± 2.5)	1.6–9.6 (5.1 ± 1.93)	1.9–9.7 (3.9 ± 1.5)	
Call period (s)	0.003–0.9 (0.2 ± 0.06)	0.002–0.7 (0.25 ± 0.09)	0.003–0.6 (0.2 ± 0.08)	
Peak frequency (Hz)	1,764–2,150 (1,934 ± 65.8)	1,620–2,882 (2,583 ± 158)	2,030–2,870 (2,421 ± 205)	
Entropy	0.09–0.21 (0.14 ± 0.02)	0.23–0.58 (0.33 ± 0.05)	0.19–0.95 (0.30 ± 0.05)	
	Within the range of Hyla orientalis	
Arkutino	Dobrusha	Plovdiv	
Call count	14–56 (38 ± 11)	12–73 (37 ± 16)	20–72 (41 ± 11)	
Pulse count	6–10 (7.8 ± 0.7)	5–10 (7.4 ± 1.12)	6–10 (7 ± 0.8)	
Call series duration (s)	2.6–14.0 (8.8 ± 2.9)	1.6–16.5 (6.8 ± 3.7)	3.14–11 (6.2 ± 1.7)	
Call period (s)	0.003–0.6 (0.2 ± 0.06)	0.003–0.37 (0.17 ± 0.05)	0.001–0.36 (0.15 ± 0.05)	
Peak frequency (Hz)	1,520–2,882 (2,565 ± 168)	1,529–2,842 (2,162 ± 215)	1,287–2,960 (2,388 ± 208)	
Entropy	0.11–0.29 (0.20 ± 0.02)	0.10–0.38 (0.20 ± 0.05)	0.13–0.77 (0.39 ± 0.13)	

The MANCOVA indicated that the effect of “Site” was significant (Pillai’s trace = 1.79422, F(4, 38,968) = 990.5, p < 0.001). LSD tests revealed statistically significant differences between call parameters across sites Arkutino, Dobrusha and Plovdiv on one side and Livada, Rupite and Ruzhdak on the other (p < 0.001).

The first two components of the PCA explained 40% and 31% of the total variance, respectively. The variables with the highest loading values for these components were Call count, Call series duration, Peak frequency and Entropy (Table 2). The PCA scatterplot revealed that sites associated with H. arborea were grouped closely together, and while sites associated with H. orientalis displayed greater variance, they still formed a distinct group (Fig. 2).

Table 2 Loadings of variables onto Principal Components, with respective eigenvalues below each component.

	Comp.1	Comp.2	Comp.3	Comp.4	Comp.5	Comp.6	
2.331	1.842	0.961	0.521	0.215	0.015	
Call count	0.629	–	–	0.339	0.144	0.684	
Pulse count	0.454	0.185	−0.101	−0.766	−0.340	–	
Call series duration	0.564	−0.179	0.379	0.268	–	−0.657	
Call period	−0.148	−0.417	0.616	–	−0.485	0.295	
Peak frequency	–	−0.662	−0.128	−0.390	0.624	–	
Entropy	–	−0.577	−0.671	0.256	−0.486	−0.104	

Figure 2 PCA scatterplot of mating calls distribution across the different study sites.

Results from the LDA revealed that based on the average distances, the sites were divided into two distinct groups: on one side Arkutino (0.783), Dobrusha (0.763) and Plovdiv (0.787), and on the other Livada (0.627), Rupite (0.628) and Ruzhdak (0.618). To better visualise the structural differences between the two groups, a call from a single location from each group was presented using a spectrogram and oscillogram (Fig. 3). As a whole, the most easily distinguishable characteristic of the calls from the first group (sites in the range of H. orientalis) were the higher values for call count and call series duration in comparison to calls from the second group (sites in the range of H. arborea). While Pulse count also was generally higher for the first group, its loading value from the PCA was lower (Table 2).

Figure 3 Spectrogram and oscillogram of calls from Ruzhdak (A) and Plovdiv (B).

Above each call group is a representation of a single call with the mean number of pulses for the respective site.

Discussion

The six sites presented in this study form two distinct groups that correspond well to the known range of the two species, which is an indicator that despite their similarity, mating calls can be used to differentiate between these taxa. The most important call parameters for this differentiation were call count, Call series duration, Peak frequency and Entropy.

Call parameters have been used as additional traits for better species assignment in anurans for the past several decades, even though the focus has primarily been on the water frogs from the Pelophylax genus. A well-known example of morphologically almost identical species with very similar calls would be the sister species of Pelophylax ridibundus, P. bedriagae and P. kurtmuelleri (Schneider & Sinsch, 1992, 1999). In the most recent update of the species list of European herpetofauna, P. kurtmuelleri is not recognised as a valid taxon, while the status of P. bedriagae is somewhat uncertain, with a wide hybrid zone suggesting European and Anatolian lineages of the Pelophylax ridibundus–bedriagae complex are conspecific (Speybroeck et al., 2020). Most recently, acoustic data was used to suggest removal of the species status of Pelophylax caralitanus (controversial taxon suggested in 2001) and its synonymization with P. bedriagae (Sinsch, Werding & Kaya, 2023).

While the two closely related species H. arborea and H. orientalis have so far only been distinguished based on molecular data, there are bioacoustics studies on Hyla species that have used recordings assigned as “Hyla arborea” from sites within the H. orientalis range (Kaya & Simmons, 1999; Schneider, 2001). Although this work provides the first direct comparison between the mating calls of H. arborea and H. orientalis, there is a substantial number of studies focused on the calling characteristics of European treefrogs. Kaya & Simmons (1999) compared calls from Izmir and Beysehir Lake (then sites for H. arborea, today in H. orientalis range) to calls from Anamur and Adana (H. savignyi), and found that call duration, intercall interval and number of pulses per call were important for differentiating between frogs in different sites. The mean number of pulses per call for “H. arborea” was 8.52, which is similar to what was registered for H. orientalis in this study. More bioacoustics data clarifying the distribution of H. arborea and H. savignyi on the south coast of Turkey is provided by Schneider (2001), with mean number of pulses per call being 8.80. Based on previous publications, Gvoždík et al. (2015) group the mating calls of Western Palearctic treefrogs into ‘long-calls’ (characterized by a high number of pulses; 34–56 pulses; 210–610 ms), ‘short-calls’ (low number of pulses; 6–12 pulses; 50–110 ms), and ‘medium-calls’ (intermediate values; 13–25 pulses; 85–190 ms). In this regard, H. arborea and H. orientalis are grouped together as “short call” species, which is also reflected in this study, as the Pulse count variable was not among the four parameters that accounted for the majority of the total variance. Even though the pulse count was slightly higher for sites with H. orientalis, this number is still below what was reported for H. arborea from Germany (9.1 ± 0.4, Schneider, 1977, 2000) and Italy (8.2 ± 0.8, Castellano et al., 2002). In this study, the two variables with the highest values for species differentiation were call count and call series duration (see Table 2). It is known that call parameters could be influenced by the individual’s condition and various environmental factors, especially temperature (Kuczynski et al., 2010; Vélez & Linehan-Skillings, 2013). However, according to Schneider (2004), call series duration, number of calls per call series and number of pulses per call were all unaffected by air temperature, which varied between 9 °C and 20 °C. It has to be noted that in Schneider (2004), these parameters are referred to as “call duration, number of pulse groups per call and number of pulses per group”, which are traditional labels and are widely used in anuran bioacoustics. Schneider (2004) also reports that temperature did affect call (pulse) duration and interval–in the present study, these parameters (combined in the variable call period) were less important for differentiating between sites. Curiously, Schneider (1967), Schneider (2004) remarks that while most call series were comprised of 15–30 calls (pulse groups), during the peak of the breeding season, this number reached 100–180 or even 244–362. These numbers are very interesting, considering that the longest call series for this study contained 73 calls, and the call series from sites with H. arborea were significantly shorter (see Table 1). Call series duration for H. arborea given by Castellano et al. (2002) was significantly longer (11.1 ± 5.3) than what is reported in the present study. This might be explained at least partially by the fact that the time an individual male spends calling can affect its call parameters. Castellano & Gamba (2011) have established that during sustained calling in Hyla intermedia, call duration tends to increase and pulse rate to decrease; however, given the size of the sample used in this study, this would be an unlikely explanation for the observed differences. Although data from the contact zone of H. orientalis and H. arborea in Bulgaria are still insufficient, it is likely that both species can occur in sympatry in some areas in the south-western parts of the country. It is possible that at least some of the registered results are due to reproductive character displacement, which has been documented for the North American species Hyla cinerea and H. gratiosa (Gordon, Ralph & Stratman, 2017; Höbel & Gerhardt, 2003). However, all sites were specifically chosen so that sympatry and syntopy (i.e., the simultaneous occurrence of both species in the same site) were extremely unlikely. The study sites for both species were based on the sampling localities provided in Dufresnes et al. (2015) and the nearest site for H. arborea was approximately 150 km away from the region in Bulgaria where the two species might live in sympatry (Osogovo Mountain). The sites for both species were hundreds of kilometres away from the nearest known hybrid zones in South-Eastern Serbia and North-Eastern Greece.

In their study on nine H. arborea populations, Castellano et al. (2002) found statistically significant negative correlation between temperature and call duration in three populations; in a single population, there was significant positive correlation between temperature and fundamental frequency. In their study on Hyla intermedia and H. sarda, Rosso, Castellano & Giacoma (2004) also establish correlation between temperature and temporal call parameters, as well as between body size and spectral parameters. All calls analysed in this study are from recordings with similar temperatures (18–20 °C), but unfortunately the effect of body size cannot be estimated; still, it is unlikely that random size differences could account for the clear grouping of the six study sites. For Rosso, Castellano & Giacoma (2004) the differences in frequency were still significant even when the effect of body size was removed, but it is also possible that males of H. orientalis are smaller and therefore call at higher frequency than males of H. arborea.

Entropy is the fourth call parameter that had significant value in group differentiation. In Avisoft, this parameter allows to quantify the pureness of sounds and theoretically is zero for pure-tone signals and one for random noise. Specifically, whistle-like sounds usually have a low entropy (<0.3), while noisy sounds have higher entropies (>0.4). Since spectrograms for all calls were generated using the same spectrogram parameter settings, comparisons between sites should be valid, however, the significance of this result is unclear, and more recordings in controlled environment should be analysed in order to draw any conclusions.

Although a large portion of existing data on treefrog calls is decades old (e.g., Schneider, 1967, 1968, 1974, 1977) it is very unlikely that recordings made at the same sites with modern technology would produce different results, as the data from these studies is consistent with newer findings.

Conclusions

The present study tested the hypothesis that the two closely related species H. arborea and H. orientalis can be distinguished based on the parameters of their mating calls. Results indicate that calls from different sites fall within two distinct groups, which correspond with the known distribution range of the two species. Spectrograms of calls from the two groups show visible differences in call group duration, call and pulse count, and the results provide first data apart from molecular evidence for the species status of these taxa. Notable limitation of this study is that calling frogs were not captured and measured, so size differences might be the reason for at least some of the observed differences in frequency. Another limitation is that, except for Livada, daily temperature was not measured on site, but it is very unlikely that the differences in call parameters are due to temperature discrepancies. The future direction for research on this topic would be to conduct behavioral experiments to test frog responses to different stimuli, ideally backed with detailed morphometric measurements from the respective populations. It would also be beneficial to compare similar databases with frog calls from other countries within the range of the species.

Supplemental Information

Supplemental Information 1 Raw data used for analyses.

All call parameters used in the article.

I would like to dedicate this study to my late mentor and colleague Nikolay Tzankov, in gratitude for his guidance, knowledge and enthusiasm. I thank Miroslav Slavchev for his assistance with the audio recordings from Dobrusha.

Additional Information and Declarations

Competing Interests

Author Contributions

Animal Ethics

Data Availability

The author declares that he has no competing interests.

Simeon Lukanov conceived and designed the experiments, performed the experiments, analyzed the data, prepared figures and/or tables, authored or reviewed drafts of the article, and approved the final draft.

The following information was supplied relating to ethical approvals (i.e., approving body and any reference numbers):

The research was carried out in accordance with Bulgarian Ministry of Environment and Water permit No 861/13.01.2021.

The following information was supplied regarding data availability:

The raw data from all analysed calls are available in the Supplemental File.

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
