# Peer review of "Not so cryptic–differences between mating calls of Hyla arborea and Hyla orientalis from Bulgaria"

_PeerJ, doi:10.7717/peerj.17574_

## Round 0.1 · original submission · Major Revisions

Their work presents very valuable information about the call of both Hyla species studied and is also the first contribution for Bulgaria based on their call qualities.
As you will see below, comments from two referees suggest a major revision before your paper can be published. Their comments should provide a clear idea for you to review, hopefully improving the clarity and rigor of the presentation of your work. I will be happy to accept your article pending further revisions, detailed by the referees, which largely focus on clarifying various aspects of your work.

Reviewer 1 made comments basically pointing out insufficiency of literature on the terminology used.

Reviewer 2 basically points out a statistical problem of pseudoreplication for male and for group, and also highlights the case of possible overlapping distribution ranges between both Hyla species, which can significantly influence the results obtained.

**Language Note:** The review process has identified that the English language must be improved. PeerJ can provide language editing services - please contact us at [email protected] for pricing (be sure to provide your manuscript number and title). Alternatively, you should make your own arrangements to improve the language quality and provide details in your response letter. – PeerJ Staff
Please note that we consider these revisions to be important and your revised manuscript will likely need to be revised again.

·

Basic reporting

The ms provides the first attempt for Bulgaria to distinguish two Hyla species based on their advertisement call features. English should be checked by a fluent speaker. Literature coverage is not sufficient (see comments in the uploaded ms pdf). Consequently, terminology used deviates from the usually adopted one without giving a reason, why there is this deviation.

Article structure and hypothesis are okay.

Experimental design

This study is not experimental but descriptive.

Validity of the findings

I do not have doubts on the validity of findings. Underlying data are presented, but seem to be selected according an unknown criterion because the number of calls analysed per locality seems low considering the four-years study period. Conclusions are well stated and are supported by the results.

Additional comments

Additional comments are written directly into the uploaded pdf. My major concerns are with terminology and literature coverage.

Reviewer 2 ·

Basic reporting

The manuscript titled “Not so cryptic - differences between mating calls of Hyla arborea and Hyla orientalis from Bulgaria ” presents call analysis of two species of treefrogs currently only identified by molecular differences, to examine potential call differences useful for species delimination. The data is valuable, but the statistical analysis if flawed (see below).

Experimental design

Major comment 1: Statistical analysis: Pseudoreplication on two levels: using more than one call group from a given male, and using more than one call per call group. Both easily fixed by including individual ID as a random term or by calculating averages from the multiple measures per male.

Major comment 2: In the methods section and based on Fig 1, the reader is led to believe that distributions are allopatric. But in the discussion the reader is told that H. arborea and H. orientalis occur in sympatry in Bulgaria. This needs to be clarified, because if the green shading in Fig. 1 indicates area of sympatry there is no way to know if the calls from South-West Bulgaria are in fact all H. arborea or a mix of the two species. In which case the comparison is useless.

Major comment 3: Besides providing one exemplar call group from each area there is little visual data presentation that shows the differences between the calls. A PCA scatterplot showing the calls from the different locations, and how much they overlap, would be very helpful.

Minor comments:

Methods:

Line 94: the author states that a total of 390 calls were analyzed and included in the analysis. However, review of the provided xls data spreadsheet shows that those were “call groups”, not “calls”, and that the total number of analyzed “calls” is ins fact n=9752 . This totally inflates sample size and is likely responsible for highly significant statistics despite very small call differences (indicated by Table 1). Also, the 9 to 73 individual calls from each call group are not independent. Either include individual caller ID as a term in the statistical model , or make averages of the data from the individual calls.

Results:
Line 138: what were the eigenvalues of the first two PCA components?
Line 140 (Table 2): Why give factor loadings of 7 components in the table when the text refers to only the first two? Did they all have eigenvalues above 1?

Figure 3: the x-axis showing the call groups are different in the two panels. X-axis in Panel A is around 6 seconds, but panel B it is around 11 seconds. This makes the call groups look much more different than they actually are. Also, considering the oscillograms showing individual calls above each call group: (1) no timing information is provided for them, and (2) they seem to be shown with similar timing (aka, they maybe more comparable than the call groups, but the info in the figure is missing).

Figures:
Given that a map indicating the locations of the study sites (Fig. 1) is provided, the LDA analysis and Fig 2 are superfluous.

Validity of the findings

no comment

---

## Round 0.2 · accepted · Accept

The reviewers and I have checked the corrections made to the manuscript, and we have seen that each of the noted observations were correctly addressed. Consequently, I am happy with this version because I consider that it already meets the necessary conditions for publication.

·

Basic reporting

The revised ms has adressed all issues raised in the earlier version. Literature coverage is now satisfactory.

Experimental design

does not apply. Methods are described with sufficient detail

Validity of the findings

Data collection and analyses are sound. Conclusions follow from the data.

Additional comments

The ms in its present state is fine. No further comments.